# Identification of preclinical dementia according to ATN classification for stratified trial recruitment: A machine learning approach

Ivan Koychev[1]*, Evgeniy Marinov[2], Simon Young[1], Sophia Lazarova[2], Denitsa Grigorova[2,3], Dean Palejev[2,4]

1 Department of Psychiatry, University of Oxford, Oxford, United Kingdom, 2 Big Data for Smart Society (GATE) Institute, Sofia University, Sofia, Bulgaria, 3 Faculty of Mathematics and Informatics, Sofia University, Sofia, Bulgaria, 4 Institute of Mathematics and Informatics, Bulgarian Academy of Sciences, Sofia, Bulgaria

* ivan.koychev@psych.ox.ac.uk

**Data Availability Statement:** As the minimal dataset was thus obtained from third parties (i.e., data not owned or collected by the authors), authors can access the specific datasets related to

## Abstract

### Introduction

The Amyloid/Tau/Neurodegeneration (ATN) framework was proposed to identify the preclinical biological state of Alzheimer's disease (AD). We investigated whether ATN phenotype can be predicted using routinely collected research cohort data.

### Methods

927 EPAD LCS cohort participants free of dementia or Mild Cognitive Impairment were separated into 5 ATN categories. We used machine learning (ML) methods to identify a set of significant features separating each neurodegeneration-related group from controls (A-T-(N)-). Random Forest and linear-kernel SVM with stratified 5-fold cross validations were used to optimize model whose performance was then tested in the ADNI database.

### Results

Our optimal results outperformed ATN cross-validated logistic regression models by between 2.2% and 8.3%. The optimal feature sets were not consistent across the 4 models with the AD pathologic change vs controls set differing the most from the rest. Because of that we have identified a subset of 10 features that yield results very close or identical to the optimal.

### Discussion

Our study demonstrates the gains offered by ML in generating ATN risk prediction over logistic regression models among pre-dementia individuals.

our research through the EPAD (https://ep-ad.org/open-access-data/access/) and ADNI (https://adni.loni.usc.edu/data-samples/access-data/) data access procedures. Researchers would be able to access or request these data in the same manner as the authors. The authors confirm that they did not have any special access or request privileges that others would not have.

**Funding:** Data collection and sharing for this project was funded by the Alzheimer's Disease Neuroimaging Initiative (ADNI) (National Institutes of Health Grant U01 AG024904) and DOD ADNI (Department of Defense award number W81XWH-12-2-0012). ADNI is funded by the National Institute on Aging, the National Institute of Biomedical Imaging and Bioengineering, and through generous contributions from the following: AbbVie, Alzheimer's Association; Alzheimer's Drug Discovery Foundation; Araclon Biotech; BioClinica, Inc.; Biogen; Bristol-Myers Squibb Company; CereSpir, Inc.; Cogstate; Eisai Inc.; Elan Pharmaceuticals, Inc.; Eli Lilly and Company; EuroImmun; F. Hoffmann-La Roche Ltd and its affiliated company Genentech, Inc.; Fujirebio; GE Healthcare; IXICO Ltd.; Janssen Alzheimer Immunotherapy Research & Development, LLC.; Johnson & Johnson Pharmaceutical Research & Development LLC.; Lumosity; Lundbeck; Merck & Co., Inc.; Meso Scale Diagnostics, LLC.; NeuroRx Research; Neurotrack Technologies; Novartis Pharmaceuticals Corporation; Pfizer Inc.; Piramal Imaging; Servier; Takeda Pharmaceutical Company; and Transition Therapeutics. The Canadian Institutes of Health Research is providing funds to support ADNI clinical sites in Canada. Private sector contributions are facilitated by the Foundation for the National Institutes of Health (www.fnih.org). The grantee organization is the Northern California Institute for Research and Education, and the study is coordinated by the Alzheimer's Therapeutic Research Institute at the University of Southern California. ADNI data are disseminated by the Laboratory for Neuro Imaging at the University of Southern California. The funders had no role in study design, data collection and analysis, decision to publish, or preparation of the manuscript.

**Competing interests:** I.K. is paid medical advisor for digital technology companies developing solutions for the early diagnosis and care of dementia (Five Lives Ltd, Cognetivity Ltd and Mantrah Ltd). This does not alter our adherence to PLOS ONE policies on sharing data and materials.

# Introduction

It is now widely recognized that dementia is a clinical syndrome that is a complication of a variety of pathophysiological processes that typically run a lengthy preclinical course. The Amyloid/Tau/Neurodegeneration (ATN) biomarker framework [1] was proposed as means of evidencing the biological state of Alzheimer's disease (AD) and non-AD pathophysiology that is independent of clinical manifestation. This novel definition of neurodegenerative states has operationalized the recent shift in interventional dementia trials from the syndromal stages of disease to biologically defined preclinical states [2].

The scalability of the ATN framework that is required for large interventional trials is limited by the availability, invasiveness and cost of the biomarker investigations. Prediction of ATN status through known risk factors [3] and dementia scores [4] may help address this limitation through empowering targeted recruitment. A substantial body of literature exists demonstrating the utility of both regression and machine learning based methods to detect amyloid positivity in cognitively healthy individuals, highlighting the role of APOE4 carriership, demographic factors [5,6] as well as neurodegeneration plasma biomarkers [7,8]. ATN status determination has the additional benefit of indicating the stage of disease (i.e. presence of neurodegeneration) as well as identifying non amyloid driven pathology. We have previously shown that ATN prediction is feasible using regression-based modelling [9]. We found that continuous constitutional and modifiable risk factors for late-life dementia outperform the best established future dementia risk scores (e.g. Cardiovascular Risk Factors, Aging and Dementia (CAIDE) [4] and Framingham risk scores [10,11]).

In this study, we sought to explore if further gains in ATN prediction could be granted by artificial intelligence in the form of machine learning (ML) modelling relative to the best regression models. The rationale for this approach is the evidence that data-driven approaches such as ML can outperform classical statistical method in the field of diagnostics. In addition, such algorithms can continuously 'learn' and improve over time as data accumulates. We approached the research question by generating the ATN ML model in the European Prevention of Alzheimer's Dementia Longitudinal Cohort Study (EPAD LCS) dataset [12] in order to compare with our previous regression model [9] and then validating it in the Alzheimer's disease Neuroimaging Initiative (ADNI) dataset [13].

# Methods

## Study design and aims

The present study extends previous work [9] that identified 7 risk factors for AD in relation to prediction of ATN pathology in EPAD. We improve the previously reported results by using Machine Learning (ML) methods instead of logistic regression. We planned to do this, applying feature selection methods aimed at identifying additional significant factors for prediction of ATN pathology. Finally, we planned to validate the selected features on independent data from ADNI.

## Study cohorts

**Training and testing dataset: EPAD LCS.** EPAD LCS is a multicenter, longitudinal cohort study recruiting participants across 21 European sites [12] and we used its V1500.0 dataset to obtain data from 1500 adult participants aged over 50 years. To replicate the previous study [9], we excluded 82 individuals due to diagnosis of dementia or MCI and additional 171 participants were excluded due to Clinical Dementia Rating (CDR) score ≥ 0.5. Further 320 participants were omitted due to missing data. The sample sizes of the different groups used in this work are identical to the sample size reported for the logistic regression analysis

reported previously [9] in the legend of Table 4 of the original publication, resulting of a total of 927 individuals (age 64.5 ± 6.77 years, 58.4% female).

**Validation dataset: ADNI.** The Alzheimer's Disease Neuroimaging Initiative (ADNI) is a multicenter longitudinal study launched in 2004. ADNI follows a population of volunteers classified as either cognitively normal (CN), living with significant memory concern (SMC), mild cognitive impairment (MCI), or Alzheimer's disease (AD). Full details and protocols can be found at http://adni.loni.usc.edu/.

Out of 3285 ADNI participants, 2578 participants had missing data and were omitted. This led to a sample of 709 individuals (72.2 ± 7.27 years, 47% female) that consisted of 109 (15.37%) AD, 143 (20.17%) CN, 367 (51.76%) MCI and 90 (12.69%) SMC individuals. As applying identical exclusion criteria to the EPAD dataset (i.e. exclusion of all MCI or dementia cases) produced a dataset of insufficient size, we opted not to apply this.

As ADNI was a validation dataset, we selected features that we found to be significant in predicting ATN pathology in EPAD. Due to the established relevance of plasma biomarkers in AD prediction [14], we also included concentrations of plasma phosphorylated tau181 (p-tau181).

**Ethics.** The EPAD and ADNI studies received appropriate ethical approval in line with the Declaration of Helsinki. Participants provided written consent prior to any study procedures.

## Study assessments

**Constitutional risk factors.** Constitutional factors included age, gender, years of education and family history of dementia. Family history of dementia was included as a binary variable showing the presence or absence of diagnosed parental dementia for ADNI and first-degree relative dementia for EPAD.

## Cardiovascular variables

Systolic blood pressure (BP) and body-mass index (BMI, derived from height and weight) were included as continuous variables for both ADNI and EPAD cohorts. Smoking score was encoded as a categorical variable with 5 levels (Table A from S1 Appendix) derived from gender, age and smoking status.

EPAD sample also includes physical activity encoded as an ordinal variable with 6 levels (Table B from S1 Appendix).

For both EPAD and ADNI cohorts the proportion of white matter lesion (WML) volume per whole brain volume was derived from volumetric MRI data. Brain imaging protocols for EPAD have been previously described [15] and further details on protocols and procedures for ADNI can be found here http://adni.loni.usc.edu/methods/documents/.

**Cognition.** Global cognitive function was derived from total Mini-Mental State Examination (MMSE) score [16]. Performance scores from Repeatable Battery for the Assessment of Neuropsychological Status (RBANS) tests were also included in the EPAD cohort [9]. For the purpose of this study our ADNI sample contained only the results from a semantic fluency assessment task that was identical to the one featured in RBANS.

**APOE4 carriership.** Genetic risk was encoded as a binary variable denoting whether a participant was an APOE4 carrier or not. An individual was designated an APOE4 carrier if they had one or more copies of allele 4. A non-carrier was considered to be an individual who had no allele 4 in their genotype. Details on apolipoprotein E (APOE ε4) genotyping methods and collection protocols for EPAD and ADNI can be found at https://ep-ad.org/ and http://adni.loni.usc.edu/, respectively.

**CSF Biomarkers.** Cerebral spinal fluid (CSF) concentrations of amyloid beta 42 (Aβ42) and phosphorylated tau (p-Tau) were included in both samples. For the ADNI sample we also included CSF concentrations of total tau (t-Tau) as a neurodegeneration marker [17]. The cut-off for definition of Aβ42 pathology (A+) and p-Tau pathology (T+) for EPAD was < 1025 pg/ml and > 24 pg/ml, respectively [9]. These cut-offs were derived by the authors using Gaussian Mixture Modeling. For ADNI we used cut-offs that were previously validated against amyloid PET visual read. The cut-offs were as follows: A(+) < 1000 pg/ml, (T+) > 27pg/ml and (N+) > 300 pg/ml ([18–20], Clinical Values sections).

**Neurodegeneration biomarker.** For the EPAD cohort neurodegeneration was estimated with the Scheltens' visual rating scale [9]. Scheltens' scale is widely used to assess medial temporal atrophy from structural MRI images [21]. Neurodegenerative pathology (N+) was defined using the following cut-offs: scores > 1 for participants less than 65 years old, > 1.5 for 65 to 74-year-olds, and > 2 for participants older than 75 years [22].

The presence of neurodegeneration in the ADNI sample was inferred from the CSF concentrations of t-Tau. Neurodegenerative pathology (N+) was defined as t-Tau > 300 pg/ml [20].

**Plasma biomarkers.** A constantly accumulating body of literature suggests that levels of p-tau181 measured in blood plasma might be useful as a biomarker for AD-related neurodegeneration [23–26]. Higher levels of p-tau181 in plasma have been shown to correlate with neurodegeneration and to even predict decline in aging and Alzheimer's disease [23,25,27]. Furthermore, measuring p-tau181 in blood plasma is much more cost-effective and less invasive than a lumbar puncture or a neuroimaging procedure.

**ATN framework.** Participants were categorized into five subgroups in accordance with the ATN Framework [1]:

i. Normal AD biomarkers: A−T−(N)−

ii. Alzheimer's pathologic change: A+T−(N)−

iii. Alzheimer's disease: A+T+(N)±

iv. Alzheimer's and concomitant non-Alzheimer's pathologic change: A+T−(N)+

v. Non-AD pathologic change: A−T ± (N)+; A−T+(N)−

## Analysis

**Feature and model selection.** We used Information Value (IV) and Weight of Evidence (WOE) [28,29] to determine the most significant features from the EPAD data set. The WOE measures the predictive ability of a feature with respect to the dependent variable. Encoded values of a variable into discrete categories are used to compute WOE assigned to each category. The formulas are shown in [28], Section 2 and the heuristics are described in [29]. The larger the absolute value of WOE, the more discriminative is the corresponding category among the values of the considered variable with respect to the dependent variable. An important assumption is that the dependent variable should be binary to denote the occurrence or no-occurrence of an event. The formulas for IV are again shown in [28], Section 2, and we have used a modified and extended version of the implementations from [30]. Based on these methods, we selected a set of 13 features, including the 7 used in [9] (female, age years, family history, APOE4, Body Mass Index (BMI), white matter lesion volume, MMSE score) and 6 that were not previously included—years of education, RBANS semantic fluency, RBANS delayed memory index, smoking score (ever smoked) Framingham I, physical activity, systolic blood pressure. Summaries of these 13 features from the EPAD and two ADNI datasets that

**Table 1. Descriptive statistics (mean±sd for continuous variables, percentage for binary) of the variables included in the models for European Prevention of Alzheimer's Dementia Longitudinal Cohort Study (EPAD) and Alzheimer's Diseasing Neuroimaging Initiative (ADNI).**

| Variable: | EPAD | | ADNI | | ADNI reduced | |
|---|---|---|---|---|---|---|
| Continuous: | | | | | | |
| Education (years) | 14.66 ± 3.63 | | 16.27 ± 2.61 | | 16.66 ± 2.47 | |
| Age (years) | 64.54 ± 6.77 | | 72.22 ± 7.27 | | 72.65 ± 6.12 | |
| BMI | 26.3 ± 4.2 | | 27.58 ± 5.28 | | 27.68 ± 5.14 | |
| MMSE total | 28.84 ± 1.33 | | 27.66 ± 2.49 | | 29.05 ± 1.21 | |
| RBANS semantic fluency | 20.46 ± 5.25 | | 18.26 ± 5.86 | | 21.22 ± 5.29 | |
| White matter lesion volume | 13.18 ± 31.83 | | 6.84 ± 9.82 | | 6.07 ± 10.94 | |
| Systolic blood pressure | 134.29 ± 17.46 | | 132.66 ± 16.99 | | 133.77 ± 16.64 | |
| Plasma p-tau181 | NA | | 18.59 ± 19.72 | | 17.04 ± 30.24 | |
| RBANS delayed memory index | 108.2 ± 59.53 | | NA | | NA | |
| | | | | | | |
| Binary: | | | | | | |
| Female | 58.36% | | 46.97% | | 52.36% | |
| APOE4 (carrier) | 38.51% | | 45.28% | | 29.18% | |
| Dementia family history (yes) | 71.95% | | 57.4% | | 53.65% | |
| Categorical: | | | | | | |
| Smoking score (ever smoked) Framingham I, % | 0 | 1 | 2 | 3 | 4 | |
| EPAD | 45.42 | 25.03 | 17.04 | 4.31 | 8.20 | |
| ADNI | 61.07 | 30.47 | 7.05 | 0.71 | 0.71 | |
| ADNI reduced | 58.37 | 34.33 | 6.87 | 0 | 0.43 | |
| Physical activity categorical, % | 0 | 1 | 2 | 3 | 4 | 5 |
| EPAD | 12.08 | 6.69 | 6.90 | 16.61 | 39.27 | 18.45 |

'ADNI' and 'ADNI reduced' refer to versions of the ADNI dataset with and without individuals with dementia or mild cognitive impairment.

Abbreviations: BMI–body mass index; MMSE–Mini-Mental State Examination; RBANS—Repeatable Battery for the Assessment of Neuropsychological Status.

we consider (see the Validation Procedure section) are shown in Table 1. All of these features with the exception of RBANS delayed memory index and physical activity were also present in the ADNI dataset.

In addition to the logistic regression considered in [9], we also applied several tree-based and kernel-based methods. Some of the tested ML methods include eXtreme Gradient Boosting (XGBoost [31,32]), and also the following methods implemented in scikit-learn Python library [33] random forest (RandomForest [34,35] and [36], Chapter 15), Extremely randomized trees (ExtraTrees, [37]), Bagging [38], Adaptive boost (AdaBoost [36,39], Chapter 10) and Support Vector Machines with different kernels ([36], Chapter 6). We use stratified 5-fold cross validation to provide a balance between the model complexity and the quality of predictions and to avoid overfitting ([36], p. 596; p. 613). Cross-validation optimizes the bias-variance tradeoff problem in ML by running the same model on different splits of the same data into training and test sets [40]. Using stratified cross-validation is needed due to the imbalanced classes in some of the comparisons.

Similarly to [9] we derive models for 4 comparisons, each classifying a specific ATN pathology against normal controls. As shown in Table C in S1 Appendix, each optimal model utilizes a different subset of the 13 most informative features as predictors. However, for utility purposes, we prefer models that have the same subset of predictor variables. Therefore, we applied

a method based on lattices in order to obtain such a minimal set of features. Through the lattice method we identified 10 features (the 13 without the following three: RBANS delayed memory index, physical activity, smoking score) as the optimal choice unifying the feature sets for all 4 models. Interestingly, these ten features are also present in the ADNI dataset.

The optimal methods and features were determined by maximizing the Area Under the Curve (AUC) [41].

To optimize the computational time and resources, we have used a two-step process. In the first step we used 1 000 iterations and have selected the 100 combinations of features and hyperparameters that have the highest AUC for each group and type of model. Only these best initial combinations were utilized during the second step with 10 000 iterations in order to find the optimal models. We apply the AUC on every fold of the 5-fold cross validation (CV). The final receiver operating characteristic (ROC) [42] curve is the mean of the separate ROC curves from each fold. To explore the variation in the estimates we also computed a 95% CI (confidence interval) for the AUC metric for each one of the 5 CV folds.

**Validation procedure.** First, we created a subset of the ADNI data using identical filtering as the original EPAD data excluding AD and MCI patients ('ADNI reduced'). It included (n = 112) controls and small number of patients with ATN pathology (n = 60 Alzheimer's pathologic change, n = 24 Alzheimer's disease, n = 37 Non-AD pathologic change). Because of the small sample size, the AUC results with and without cross validation were not reliable and therefore could not be used to validate the EPAD-based model. Because of this, we validated the models using the larger ADNI dataset with 709 individuals that did not exclude individuals diagnosed with AD or MCI. As there were significant differences between the distributions of the features in the EPAD and ADNI datasets, we could not use the EPAD-derived models on ADNI data directly. Instead, we validated our feature selection by selecting the optimal EPAD models features and hyper-parameters (e.g. number of estimators, maximal depth of tree, minimal number of samples required to split an internal node, minimal impurity decrease) to train the models on the ADNI data using stratified 5-fold cross-validation.

## Results

According to the ATN-defined criteria all participants were distributed in one of five groups: Normal AD biomarkers, Alzheimer's pathologic change, Alzheimer's disease, Alzheimer's and concomitant non-Alzheimer's pathologic change and non-AD pathologic change. The final distribution of both EPAD and ADNI participants is shown in *Table 2*.

We identified two tree-based methods, random forest and XGBoost, that were consistently maximizing the AUC for the EPAD data, with random forest performing better in most cases. When subsequently analyzing the ADNI data, we found that linear-kernel SVM outperforms both the logistic regression and in almost all cases random forest and XGBoost. Logistic regression work published previously [9] did not use cross validation and therefore we calculated the AUC for logistic regression with cross validation using the original 7 parameters and

**Table 2. Distribution of participants according to the ATN criteria.**

| ATN group | Primary set (EPAD) | Validation set (ADNI) |
|---|---|---|
| *Normal AD biomarkers* | 56.63% (n = 525) | 34.83% (n = 247) |
| *Alzheimer's pathologic change* | 20.17% (n = 187) | 26.37% (n = 187) |
| *Alzheimer's disease* | 6.90% (n = 64) | 28.34% (n = 201) |
| *Alzheimer's and concomitant non-Alzheimer's pathologic change* | 3.12% (n = 29) | 0.28% (n = 2) |
| *Non-AD pathologic change* | 13.16% (n = 122) | 10.15% (n = 72) |

compared them, together with the original results without cross validation to our results that use cross validation as typical for ML-based methods. For the EPAD data, we were able to either outperform or achieve similar results to the logistic regression with cross validation. In most cases we were even able to outperform or match the logistic regression results without cross validation. The improvements ranged between 2.2% to 8.3% in absolute terms as shown in Table 3 and Figs 1 and 2. We should note that the 2.2% increase represents almost 20% of the gap to the theoretically possible maximum. The results for XGBoost are not shown, as they were similar to those for random forest.

Although there are no universal AUC thresholds for determining the quality of a binary classification, the original logistic regression with 7 features for the Alzheimer's disease vs normal AD biomarkers groups comparison yields an AUC that is very close to the top of the "excellent" discrimination band as defined previously [43]. Nevertheless, with the original 7 features we improved the already well-performing AUC scores for both random forest and linear-kernel SVM, when using cross validation. Adding additional features resulted in AUC scores for the same comparison being in the highest band, labeled "outstanding" [43].

Using the original 7 features with cross-validation we improved by about 4.5% the comparisons between AD and non-AD pathologic change, and non-AD pathologic change on one hand and normal AD biomarkers on the other. In these cases, the logistic regression results were within the "poor" discrimination band [43] and we were able to achieve results that are well within or at least practically within the "acceptable" bracket [43]. When comparing Alzheimer's pathologic change to Normal AD biomarkers using random forest we were able to achieve similar results to the logistic regression when used with cross validation. Adding additional 6 features to the random forest models led to extra improvements between approximately 0.8% and 3.5% in AUC.

For the Alzheimer's pathologic change vs Normal AD biomarkers adding the additional 6 features and utilizing random forest resulted in an improvement of about 2.2% compared to the original cross-validated logistic regression. Overall, the AUC for the resulting set of 13 features outperformed the scores for the original logistic regression with 7 features for all comparisons, with differences being between 2.2% and 8.3%. By design, the results of the reduced set of 10 features, were similar to those of 13 features, therefore making the set of 10 features a good candidate for practical applications.

As we have used the ADNI dataset as a validation one, we are interested in comparing the best ML-based results for the 10-feature models for both datasets. The results when comparing either Alzheimer's disease or Non-AD pathologic change to Normal AD biomarkers were very similar for EPAD and ADNI. When comparing Alzheimer's pathologic change to Normal AD biomarkers, the AUC scores for the ADNI logistic regression and linear SVM models even outperformed the respective ones for EPAD by 11%. This validates the feature selection using the EPAD dataset to derive models on the ADNI dataset. Adding plasma p-tau181 yields an increase of almost 3% between the best ML models when comparing Alzheimer's disease to Normal AD biomarkers without it and an improvement of 1.7% of the ML-based 10-feature result for EPAD. It also adds over 1.4% of improvement for the Non-AD pathologic change to Normal AD biomarkers comparison. Figs 1 and 2 show ROC curves and AUC results for the four comparisons using different features and datasets.

## Discussion

In this analysis, we found that machine learning offers an advantage over logistic regression when predicting Alzheimer's and non-Alzheimer's pathology as defined through the ATN classification in pre-dementia individuals. While gains of 1–2% were evident for Alzheimer's

**Table 3. AUC metric for the corresponding groups and classifiers based on the original 7 features, the selected 10 optimal minimal subset of features and all 13 selected significant features.**

| AUC | Logistic regression, CV (no CV) | Random Forest, CV | Support Vector Machine, CV |
|---|---|---|---|
| **EPAD** | | | |
| **Original 7 features** | | | |
| 1.Alzheimer's pathologic change /187/ | 0.6355 (0.66) | 0.6311 (*) | 0.6388 (*) |
| 2.Alzheimer's disease /64/ | 0.8862 (0.89) | 0.8916 (**) | 0.8905 (**) |
| 3.AD and non-AD pathologic change /29/ | 0.6594 (0.68) | 0.7069 (***) | 0.6615 (*) |
| 4.Non-AD pathologic change /122/ | 0.6472 (0.66) | 0.6918 (***) | 0.6464 (*) |
| **Subset of 13 features** | | | |
| 1.Alzheimer's pathologic change /187/ | 0.6394 (*) | 0.6571 (**) | 0.6388 (*) |
| 2.Alzheimer's disease /64/ | 0.8945 (**) | 0.9098 (***) | 0.8959 (***) |
| 3.AD and non-AD pathologic change /29/ | 0.6928 (***) | 0.7423 (***) | 0.6899 (**) |
| 4.Non-AD pathologic change /122/ | 0.6473 (*) | 0.6998 (***) | 0.6483 (*) |
| **Subset of 10 features** | | | |
| 1.Alzheimer's pathologic change /187/ | 0.6355 (*) | 0. 6571(**) | 0.6388 (*) |
| 2.Alzheimer's disease /64/ | 0.8905 (**) | 0.9073 (***) | 0.8937 (***) |
| 3.AD and non-AD pathologic change /29/ | 0.6928 (***) | 0.7423 (***) | 0.6731 |
| 4.Non-AD pathologic change /122/ | 0.6473 (*) | 0.6918 (***) | 0.6483 (*) |
| ADNI | | | |
| Best 10 features ADNI | | | |
| 1.Alzheimer's pathologic change /187/ | 0.7414 | 0.7108 (iii) | 0.7505 (iii) |
| 2.Alzheimer's disease /201/ | 0.8939 | 0.8901 | 0.8948 (ii) |
| 3.AD and non-AD pathologic change /2/ | | | |
| 4.Non-AD pathologic change /72/ | 0.6617 | 0.6589 | 0.6941 (ii) |
| Best 10 features ADNI + plasma p-tau181 | | | |
| 1.Alzheimer's pathologic change /187/ | 0.7442 | 0.7108 (iii) | 0.7509 (iii) |
| 2.Alzheimer's disease /201/ | 0.9120 | 0.9247 (iii) | 0.9091 (ii) |
| 3.AD and non-AD pathologic change /2/ | | | |
| 4.Non-AD pathologic change /72/ | 0.6620 | 0.6904 (ii) | 0.7057 (iii) |

For the EPAD dataset

(*) indicates similar result to the one from logistic regression with 7 features with cross validation, but worse than without cross validation

(**) similar result to the logistic regression without cross validation and outperforming the logistic regression with cross validation

(***) outperforming logistic regression with and without cross validation. In some cases, the results for the same methods with different number of features are identical, this is because the smaller set optimizes the AUC for the larger too. For the ADNI dataset (ii) indicates similar result and (iii) indicates better result when compared to the best EPAD ML-based result using 10 features for the respective comparison. All comparisons were done in terms of AUC, when comparing the results up to the second digit after the decimal point, with the results rounded up to the fourth digit after the decimal point.

Abbreviations: AUC–Area under the curve; CV–cross validation.

disease and Alzheimer's pathologic change, we found that the improvement in performance was most pronounced for isolated non-Alzheimer's disease pathologic change (i.e. tau and/or neurodegeneration change in the context of normal amyloid) or concomitant Alzheimer's and non-Alzheimer's pathology states (defined as abnormal amyloid and neurodegeneration biomarkers in the absence of tau change).

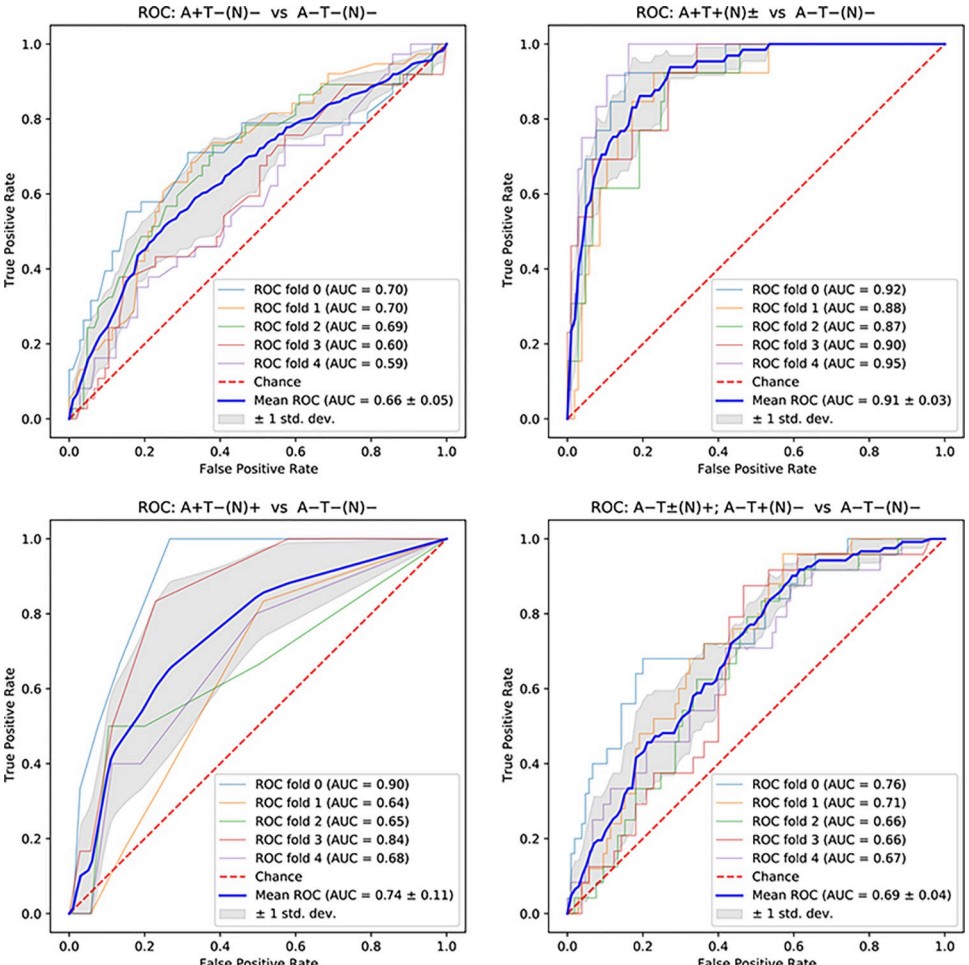

**Fig 1. ROC curves and AUC metrics for EPAD dataset using 10 features (female, age years, family history, APOE4, BMI, white matter lesion volume, MMSE score, years of education, RBANS semantic fluency, systolic blood pressure).** Each pane shows one comparison indicated on the top of the pane.

The ATN framework was developed to provide an unbiased descriptive approach to defining neurodegenerative states based on biomarker evidence that transcends the syndromal stages of the disease [44]. This allowed a reformulation of the development of disease modification agents towards individuals in preclinical disease. The biomarker-based approach allows for trials to not only test individuals with confirmed target pathology but also to modify the disease process before irreversible neurodegeneration sets in [2]. The advantages conferred by the ATN framework are however limited by the reliance on costly and invasive biomarker data being available in individuals with no cognitive change. Accurate prediction of ATN status through routinely available information is therefore a priority. In line with a previous analysis [9], we found that both Alzheimer's disease and Alzheimer's pathologic change can be predicted with high degree of accuracy (AUC of 0.89 and 0.66 respectively) through a combination of 7 variables (age, sex, APOE4 carriership, family history of dementia, BMI, MMSE score and white matter lesion hyperintensities). In the currently reported analysis, we show that ML method are not only equivalent to a cross validated version of the logistic regression but also improve on the results when an expanded dataset is used (7 original variables plus education, semantic fluency, delayed recall, smoking, physical activity and blood pressure). The validation

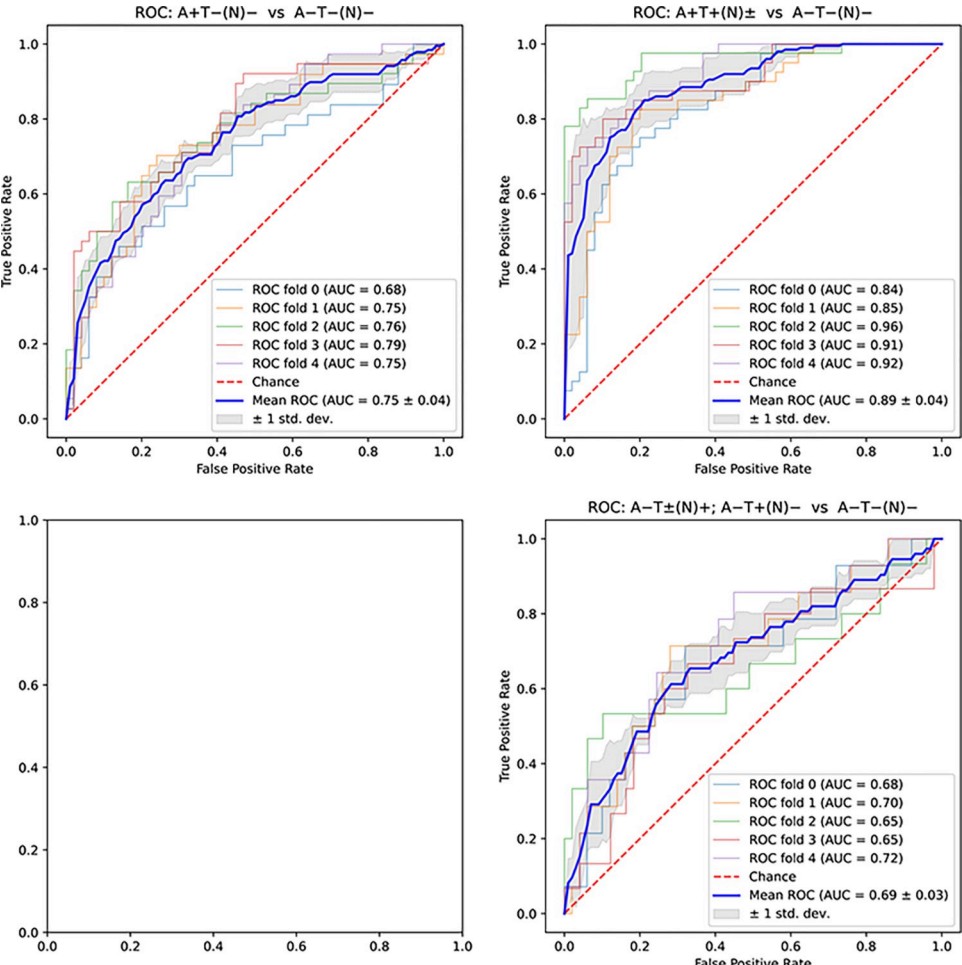

**Fig 2. ROC curves and AUC metrics for ADNI dataset using 10 features (female, age years, family history, APOE4, BMI, white matter lesion volume, MMSE score, years of education, RBANS semantic fluency, systolic blood pressure).** Each pane shows one comparison indicated on the top of the pane. The bottom-left pane is empty due to insufficient sample size.

of the 10 features ML model in ADNI demonstrated the stability of the model through AUCs of 0.89 and 0.75 for AD and AD pathological change respectively. These results are in line with other recent publications which have shown that amyloid positivity (consistent with Alzheimer's disease and Alzheimer's pathologic change) can be effectively predicted. In a recent analysis from the BioFINDER cohort, amyloid positivity in cognitively unimpaired individuals was successfully predicted through a model combining plasma Ab42/40, APOE4 carriership and age [45]. In addition, plasma biomarkers such as p-tau181 have a now well-evidenced utility in identifying AD pathology across the disease spectrum [6]. ML has also been demonstrated to have the capability to improve amyloid prediction models in cognitively unimpaired: a report from the A4 study showed that a ML model combining age, demographics, cognition and APOE4 carriership reached an AUC of 0.73 [6]. While the degree of improvement of using ML over statistical methods can be considered modest in absolute terms for the already well-optimized AD discrimination it bridges almost 20% of the gap to the theoretically possible maximum, and thus the class of ML models holds significant advantages over standard statistical methods. As shown above, the significant features are not necessarily consistent over the

different comparisons and utilizing ML methods allows us to include dependent variables without overfitting. ML methods such as random forest do not necessarily require linear separation of the categories and are typically more robust in terms of assumptions for the features distributions and allow greater flexibility.

We found that ML methods had a particularly pronounced superiority over logistic regression when predicting non-AD pathology (AUC improvement of 5% for non-AD and mixed changes in the 10 feature models). This substantial improvement likely reflects that unlike Alzheimer's pathology where the pathology is well defined, suspected non-Alzheimer's disease pathophysiology (SNAP [46]) in pre-dementia individuals is likely underpinned by a heterogenous group of pathophysiological processes, including cerebrovascular disease, Lewy body pathology, primary age-related tauopathy and argyrophilic grain disease. This lack of clear unifying features favours data driven methods such as ML. We found that non-AD pathologic change can be predicted through ML with AUCs of up to 0.69 in both EPAD and ADNI which opens the route for research of this group of disorders in their pre-dementia stages. The performance of the ML models in the mixed pathology group was similarly improved by ML in EPAD but the low number of individuals in this group (27 in EPAD and 2 in ADNI) limit the utility and performance of these algorithms.

The ADNI dataset allowed for the exploration of the added value that plasma p-tau181 offers in prediction of pre-dementia ATN status. This biomarker is one of the best validated predictors of amyloid pathology in individuals with MCI and dementia with some evidence for its utility in preclinical disease. We found that adding p-tau181 improved marginally the prediction of Alzheimer's disease and non-AD pathologic change but not the Alzheimer's pathologic change. These data add to the evidence that p-tau181 is relevant to the neurodegeneration stage of disease [14]. Other biomarkers in development such as p-tau217, p-tau231 or GFAP show promise in the preclinical stages of the disease [47] and thus may potentially offer more value in AD pathologic change stages.

## Limitations

Our study has several limitations. Firstly, in contrast to the EPAD dataset, our ADNI sample included individuals that were diagnosed with AD or MCI. The reason for this was the limited availability of ATN positive individuals who were pre-diagnosis. This resulted in different distributions of the model features for the two datasets. Because of that we performed feature validation, rather than the commonly used model validation. Future work will need to focus on further validating the models we obtained in a pre-diagnosis sample.

As our primary and validation samples were drawn from different studies, in some instance different methodologies were used to assess the same variable. For example, for EPAD we were able to use medial temporal lobe atrophy score (Scheltens' scale) to assess the level of neurodegeneration while in ADNI we used the concentrations of t-Tau in CSF. While these different means of defining neurodegeneration are recognized, this approach may have still impacted model performance.

On a similar note, some of the variables present in both data samples exhibited significant differences in their distributions. While some distributional variances were expected, WML volumes in EPAD and ADNI had particularly notable different distributions which may be accounted for by different MRI processing methodologies employed in the two studies. Q-Q plots showing these differences are shown in Supplementary Material. Despite these differences, WML variable was a consistent feature in the EPAD model when validated in ADNI.

We also observe that there is no single class of methods that reliably optimizes the comparisons for both datasets. It appears that the ADNI data is better linearly separated than the

EPAD one and because of that SVM with linear kernel (shown to be similar to logistic regression [48]) performs better on it even when adding the observations with AD and MCI. In contrast, there is no such separation in the EPAD data and because of that random forest performs better than SVM.

Finally, we note that ethnicity may conceivably be a risk factor that can be utilized in ATN prediction. However, the EPAD dataset that this analysis as well as the preceding regression analysis were based on consists almost exclusively of Caucasian individuals (785 out of the 791 individuals for which that information was available, with no information available for the remaining 136 individuals). For this reason we were not able to include this factor in our analysis. Although the ADNI sample is marginally more diverse whereby more than 92% of the individuals in that sample are Caucasian. This limitation further underlines the need for research cohorts to diversify to achieve representativeness.

## Future directions

Our work is being expanded by the development a pragmatic algorithm which allows investigators recruiting from brain health volunteer registers such as the Great Minds register [49] to select on the basis of likelihood of being in a specific ATN pathological state. In addition to accelerating preclinical dementia research, this application will allow the generation of further data on the accuracy of the models at the point of trial recruitment which will help improve the algorithm further. In addition, such approaches can, over time, yield data on the translation of ATN prediction into clinical progression over time.

A further future direction is the incorporation into the model of scalable means of AD risk monitoring such as blood biomarkers [14] and digital technology [50]. These data either add biological signal directly relevant to the core pathology [51] or have a high degree of granularity that can detect subtle changes in cognitive function [52].

## Conclusion

ML methods offer an opportunity to detect both AD and non-AD pathology through routinely collected research data. Such algorithms can be used to accelerate research into preclinical dementia states through their application in brain health volunteer registers.

## Supporting information

**S1 Appendix.**
(DOCX)

## Acknowledgments

Data used in preparation of this article were obtained from the Alzheimer's Disease Neuroimaging Initiative (ADNI) database (adni.loni.usc.edu). As such, the investigators within the ADNI contributed to the design and implementation of ADNI and/or provided data but did not participate in analysis or writing of this report. A complete listing of ADNI investigators can be found at: http://adni.loni.usc.edu/wp-content/uploads/how_to_apply/ADNI_Acknowledgement_List.pdf. The extensive numerical calculations were done on the Avitohol supercomputer that is described in [53].

A complete list of EPAD Investigators can be found at: http://ep-ad.org/wp-content/uploads/2020/12/202010_List-of-epadistas.pdf.

A complete listing of ADNI investigators can be found at: http://adni.loni.usc.edu/wp-content/uploads/how_to_apply/ADNI_Acknowledgement_List.pdf.

## Author Contributions

**Conceptualization:** Ivan Koychev, Dean Palejev.

**Data curation:** Sophia Lazarova, Denitsa Grigorova.

**Formal analysis:** Evgeniy Marinov.

**Methodology:** Dean Palejev.

**Project administration:** Ivan Koychev, Dean Palejev.

**Software:** Evgeniy Marinov, Dean Palejev.

**Supervision:** Ivan Koychev, Dean Palejev.

**Visualization:** Evgeniy Marinov.

**Writing – original draft:** Ivan Koychev, Evgeniy Marinov, Simon Young, Sophia Lazarova, Denitsa Grigorova, Dean Palejev.

**Writing – review & editing:** Ivan Koychev, Evgeniy Marinov, Simon Young, Sophia Lazarova, Denitsa Grigorova, Dean Palejev.

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
