## [Decision Letter · Decision Letter 0]

17 Dec 2022

PONE-D-22-31142Identification of preclinical dementia according to ATN classification for stratified trial recruitment: A machine learning approachPLOS ONE

Dear Dr. Koychev,

Thank you for submitting your manuscript to PLOS ONE. After careful consideration by 2 Reviewers and an Academic Editor, all of the critiques of both Reviewers must be addressed in detail in a revision to determine publication status. If you are prepared to undertake the work required, I would be pleased to reconsider my decision, but revision of the original submission without directly addressing the critiques of the Reviewers does not guarantee acceptance for publication in PLOS ONE. If the authors do not feel that the queries can be addressed, please consider submitting to another publication medium. A revised submission will be sent out for re-review. The authors are urged to have the manuscript given a hard copyedit for syntax and grammar as this is requisite for publication consideration.

We look forward to receiving your revised manuscript.

Kind regards,

Stephen D. Ginsberg, Ph.D.

Section Editor

PLOS ONE

Journal Requirements:

"I.K. is paid medical advisor for digital technology companies developing solutions for the early diagnosis and care of dementia (Five Lives Ltd, Cognetivity Ltd and Mantrah Ltd)."

**Comments to the Author**

1. Is the manuscript technically sound, and do the data support the conclusions?

Reviewer #1: Yes

Reviewer #2: No

2. Has the statistical analysis been performed appropriately and rigorously? 

Reviewer #1: Yes

Reviewer #2: Yes

3. Have the authors made all data underlying the findings in their manuscript fully available?

Reviewer #1: Yes

Reviewer #2: Yes

4. Is the manuscript presented in an intelligible fashion and written in standard English?

Reviewer #1: Yes

Reviewer #2: Yes

5. Review Comments to the Author

Reviewer #1: 1. Is the manuscript technically sound, and do the data support the conclusions?

The manuscript appropriate using techniques and draws conclusions from results.

2. Has the statistical analysis been performed appropriately and rigorously?

Statistical and ML analysis was appropriately used. Additional metrics could have been calculated and reported to provide additional insight but the authors did explain they were focusing on AUC. Also, additional techniques could have been used, such as methods specific to unbalanced datasets.

3. Have the authors made all data underlying the findings in their manuscript fully available?

Data used was from EPAD and ADNI that can be requested by researchers.

4. Is the manuscript presented in an intelligible fashion and written in standard English?

I have no concerns regarding writing.

***See attached additional comments.***

Reviewer #2: In this study, Koychev et al used the machine learning in the European Prevention of Alzheimer’s Dementia Longitudinal Cohort Study to identify features that could discriminated neurodegenerative groups from controls and validate in in the Alzheimer’s disease Neuroimaging Initiative (ADNI) dataset. Please see below comments that need to be addressed.

1. In this study, machine learning algorithms for ATN classification only slightly outperformed the logistic regression model (2.2% to 8.3%). What is the effect size?

2. In 275-276 lines, “Adding additional 6 features to the random forest models led to extra improvements between approximately 0.8% and 3.5% in AUC”, whether the addition of 6 characteristics, which only improved the model by 0.8% and 3.5%, will lead to model overfitting.

3. Why not apply nonlinear support vector machine multilayer perceptron to compare their prediction ability,

4. What the operation used in each iteration to selected the combinations of features and hyperparameters for each group and type of model.

6. PLOS authors have the option to publish the peer review history of their article (what does this mean?). If published, this will include your full peer review and any attached files.

**Do you want your identity to be public for this peer review?** For information about this choice, including consent withdrawal, please see our Privacy Policy.

Reviewer #1: No

Reviewer #2: No

---

## [Author Response · Author response to Decision Letter 0]

23 Feb 2023

Reviewer1: 

1. Is the manuscript technically sound, and do the data support the conclusions?

The manuscript appropriate using techniques and draws conclusions from results.

Author comments: Thank you. 

2. Has the statistical analysis been performed appropriately and rigorously?

Statistical and ML analysis was appropriately used. Additional metrics could have been calculated and reported to provide additional insight but the authors did explain they were focusing on AUC. Also, additional techniques could have been used, such as methods specific to unbalanced datasets.

Author comments: Many thanks. We agree further metrics could have been used to provide further details but we believe that the current level of granularity is appropriate for a paper targeting non-ML specialists. 

3. Have the authors made all data underlying the findings in their manuscript fully available?

Data used was from EPAD and ADNI that can be requested by researchers.

Author comments: This is correct – EPAD and ADNI are freely accessible to researchers. 

4. Is the manuscript presented in an intelligible fashion and written in standard English?

I have no concerns regarding writing.

Author comments: Thank you. 

Reviewer 2:

1. In this study, machine learning algorithms for ATN classification only slightly outperformed the logistic regression model (2.2% to 8.3%). What is the effect size?

Author comments: In our work we applied ML methods for 4 binary classifications and therefore we used the industry-standard AUC metric to demonstrate the effectiveness of our methods, moreover AUC was also used in the original paper (Calvin et.al. [7]). Effect size is typically used to measure the effectiveness of the tests in classical statistical analyses, some examples being Cohen’s d (in the case of t-test), Cohen’s h (in the case of test of proportions) and odds ratio (in the case of logistic regression). However, these are not applicable for binary classifications and in such cases the diagnostic odds ratio (DOR) is typically used instead. Another advantage of AUC is that it shows the overall quality of the model, and DOR can only be calculated given a threshold for separating the two classes. We have not included DOR in our work as it is not well defined for our comparisons, because it is equal to infinity, as for the optimal separation thresholds there are zero False Positives or False Negatives (that are in the denominator of DOR). This is either due to the methods performing really well (as in the case of the most important comparison, Alzheimer’s pathologic change vs Normal AD biomarkers) or the test sets being too small, resulting again in zeroes in the off-main-diagonal elements (FN, FP) of the confusion matrices and thus zero denominators. Therefore, standard metric used to evaluate the model quality in the case of binomial classification is not well defined and because of that we have opted to use the industry-standard AUC.

While we appreciate that numerically the increase in performance may seem marginal, we would like to point out that there is very little practical room for improvement on the previous results. For example, for the most important comparison, Alzheimer's pathologic change vs Normal AD biomarkers, we mention (lines 264-265) that the original results have “already well-performing AUC scores” at about 89%. Improving to about 91% (as we mention in lines 321-322) “bridges almost 20% of the gap to the theoretically possible maximum” of 100%. Furthermore, AUC scores in the mid-90% are not feasible for real-world datasets of this size, as we are reporting on a complex condition (Alzheimer’s disease). In contrast, AUC scores of mid-90% appear only in much larger datasets, or in well-curated or generated datasets used for teaching introductory ML. As such, the theoretical way to further marginally improve the AUC scores towards the mid-90% is to have larger dataset (line 372), or to have other features available in the dataset (lines 343–347, 379-381). 

2. In 275-276 lines, “Adding additional 6 features to the random forest models led to extra improvements between approximately 0.8% and 3.5% in AUC”, whether the addition of 6 characteristics, which only improved the model by 0.8% and 3.5%, will lead to model overfitting.

Author comments: We would like to respectfully point out that overfitting is not a consideration in a well-designed ML analysis. In the original version of the paper we had a relevant statement about the applicability of overfitting concerns (lines 185-188), but did not include a reference. We have now rectified this, referring toto the seminal ESL book [34] immediately after that statement and we have added the relevant page numbers for extra clarification.

In principle, overfitting may be an issue when the same data is used for both model creation and evaluation and in such cases it would become apparent when the model is fitted to another dataset. The ML process includes a safeguard against overfitting by using k-fold cross-validation. Specifically, each of the k models are derived on a training set and evaluated on a test set that is non-overlapping with the respective training set. The results (e.g., AUC scores) are reported on the test sets, rather than the training sets. This process ensures the lack of overfitted results by design. 

3. Why not apply nonlinear support vector machine multilayer perceptron to compare their prediction ability,

Author comments: : We tried SVM with various kernels (lines 184-185), including non-linear ones. In the article show that the linear kernel outperformed the non-linear ones. There was a good linear separation between the classes and therefore we decided against all possible non-linear SVM methods. But most importantly, the non-linear SVM multilayer perceptron would derive unstable models through the large ratio of number of parameters to dataset size. That is also the reason why we did not try methods based on neural networks (also perceptron-based methods).

4. What the operation used in each iteration to selected the combinations of features and hyperparameters for each group and type of model

Author comments: We deployed a rather standard ML approach. The most informative features (that were subsequently included in the models) were selected using Information Value and Weight of Evidence (line 164). The hyperparameters were selected using a two-step process (lines 200-203), grid search on the parameter space with 1000 iterations, and then increasing the iterations to 10000 when using only the top 100 combinations of hyperparameters.

---

## [Decision Letter · Decision Letter 1]

12 Mar 2023

PONE-D-22-31142R1Identification of preclinical dementia according to ATN classification for stratified trial recruitment: A machine learning approach

PLOS ONE

Dear Dr. Koychev,

Thank you for resubmitting your work to PLOS ONE. Please make the corrections posed by Reviewer #1 so I can render a decision on this manuscript.

We look forward to receiving your revised manuscript.

Kind regards,

Stephen D. Ginsberg, Ph.D.

Section Editor

PLOS ONE

**Comments to the Author**

1. If the authors have adequately addressed your comments raised in a previous round of review and you feel that this manuscript is now acceptable for publication, you may indicate that here to bypass the “Comments to the Author” section, enter your conflict of interest statement in the “Confidential to Editor” section, and submit your "Accept" recommendation.

Reviewer #1: (No Response)

2. Is the manuscript technically sound, and do the data support the conclusions?

Reviewer #1: Yes

3. Has the statistical analysis been performed appropriately and rigorously? 

Reviewer #1: Yes

4. Have the authors made all data underlying the findings in their manuscript fully available?

Reviewer #1: Yes

5. Is the manuscript presented in an intelligible fashion and written in standard English?

Reviewer #1: Yes

6. Review Comments to the Author

Reviewer #1: In the previous round of review my comments were attached in an accompanying attached document and the review directed the authors to it saying "see attached additional comment." Unfortunately, none of the comments in this document were addressed by the authors. Please address these comments.

Previous unaddressed comments are now pasted below for your convenience:

Thanks for the opportunity to review your paper.

This is an interesting paper that looks that builds on their previous work by replacing logistic regression with machine learning methods and applying feature selection methods to identify preclinical AD state based on the ATN framework/biomarkers. Results were externally validated using ADNI data.

Comments:

ABSTRACT

o Methods: “Using machine learning (ML) we identified 13 significant features separating each 30 neurodegeneration-related group from controls (A-T-(N)-)” – This is partially a result as well as a method.

INTRODUCTION

o Line 45: “was proposed”?

o This Introduction seems to be very focused on the authors’ previous work with regression modelling as well as the potential of such methods to help targeted recruitment via, as described in the manuscript, “internet-based brain health volunteer research registers such as the Brain Health Registry [5] or the Dementias Platform UK Great Minds register.” These are specific applications related to future directions later described. Including broader context of other work that has been done would improve the Introduction.

METHODS

o Study design and aims: some of this section feels more like results than aims/design such as “by applying feature selection methods we 75 identified additional significant factors”. I this could be improved by reframing as actual aims and design.

o Lines 84-86: Here the sample size is 927, but in your previous work (citation #7) is N=1010. Can you explain the discrepancy and what you mean by the sample sizes “matching”.

o Line 163: You describe WOE. It would be helpful to also describe IV.

o Line 166-167: Can you provide more clarification for the reader for how IV and WOE are calculated other than simply referencing 26,27,28?

o Constitutional factors: Did you consider race? I imagine this was a mostly White sample. Can you please report the racial makeup of the sample used? This could be considered another limitation of the study.

o Line 183: Does the second method (following RandomForest) have a name or can be describe in addition to the citation?

o Line 189: You explain that you derive 4 models comparing ATN groups vs controls. Is there a reason that you did this rather than use a multiclass classifier?

o Line 201: Please clarify what you mean by “best”

RESULTS

o Most of the comparisons/models are performed on unbalanced samples and AUCs are often sensitive to class imbalance (especially in the case of normal vs AD & concomitant non-AD changes—in that case 525 vs 29). Was this considered during cross-validation (using, for example, stratified cross-validation)?

• Discussion/Limitations/Future Directions/Conclusion

o The Discussion does a fine job of summarizing and interpreting results from the work of the authors, but it does not contextualize the results within the large amount of work that has been and is being done both other researchers in the field. I think this needs a significant update.

o In Validation Procedures subsection as well as the Limitations, I think it needs to be more clearly pronounced that the validation is not EPAD-derived models used on ADNI, but features selected on EPAD were used in the training of models on ADNI.

FIGURES AND TABLES

o Table 1: Could you statistically compare and report p-values for differences between, say, EPAD and ADNI as well as EPAD and ADNI-reduced? It appears there are differences in the sex, family history, APOE4, etc. makeup of the samples, but I am wondering if they are statistically significant.

o Table 3: It would be helpful if you could clarify how you quantitatively compared models and determined which models were similar or outperformed other models.

o Figures 1 and 2 are nice.

CITATIONS

o Citations 16-18 are not informative enough for someone to know where to find the CSF cutoffs they reference.

Thank you for the opportunity to read your very interesting manuscript and for the contribution your work will give to the field of Alzheimer’s disease research. It was a pleasure to read your work.

7. PLOS authors have the option to publish the peer review history of their article (what does this mean?). If published, this will include your full peer review and any attached files.

**Do you want your identity to be public for this peer review?** For information about this choice, including consent withdrawal, please see our Privacy Policy.

Reviewer #1: No

---

## [Author Response · Author response to Decision Letter 1]

31 May 2023

Reviewer 1: Thanks for the opportunity to review your paper.

This is an interesting paper that looks that builds on their previous work by replacing logistic regression with machine learning methods and applying feature selection methods to identify preclinical AD state based on the ATN framework/biomarkers. Results were externally validated using ADNI data.

Comments:

ABSTRACT

o Methods: “Using machine learning (ML) we identified 13 significant features separating each 30 neurodegeneration-related group from controls (A-T-(N)-)” – This is partially a result as well as a method.

Authors’ response: Indeed, this sentence contained some of our results. We rephrased it to contain only the class of methods and their purpose.

INTRODUCTION

o Line 45: “was proposed”?

Authors’ response: This has been corrected now.

o This Introduction seems to be very focused on the authors’ previous work with regression modelling as well as the potential of such methods to help targeted recruitment via, as described in the manuscript, “internet-based brain health volunteer research registers such as the Brain Health Registry [5] or the Dementias Platform UK Great Minds register.” These are specific applications related to future directions later described. Including broader context of other work that has been done would improve the Introduction.

Answer: The reason why we focused on our existing work and the related applications is because the aim of the study was to improve specifically our previous analysis which was based on regression. We have amended the introduction to give a succinct overview of the amyloid prediction literature in particular. 

METHODS

o Study design and aims: some of this section feels more like results than aims/design such as “by applying feature selection methods we 75 identified additional significant factors”. I this could be improved by reframing as actual aims and design.

Authors’ response: We have amended the Study Design and Aims section to reflect that it referred to our planned analysis. 

o Lines 84-86: Here the sample size is 927, but in your previous work (citation #7) is N=1010. Can you explain the discrepancy and what you mean by the sample sizes “matching”.

Answer: Indeed, the original paper mentions 1010 as sample size and does not reference that it is reduced to 927 explicitly. However the corresponding Table 4 shows the actual sample sizes of the groups that were used for the analysis; the reason for the lower sample size was the extent of missing data. The group sample sizes are identical to this paper and add up to 927. “Matching” was not referring to a statistical procedure, we have replaced it with “identical”.

o Line 163: You describe WOE. It would be helpful to also describe IV.

Authors’ response: Please see the answer of the following suggestion.

o Line 166-167: Can you provide more clarification for the reader for how IV and WOE are calculated other than simply referencing 26,27,28?

Authors’ response: We have added more explanations about the design and heuristics of IV and WoE, separated the citations for the two procedures and have added the section of the references that discusses them. However, we feel that going into much deeper discussion would be out of scope of this paper, moreover these procedures are well known and well described in the references.

o Constitutional factors: Did you consider race? I imagine this was a mostly White sample. Can you please report the racial makeup of the sample used? This could be considered another limitation of the study.

Authors’ response: The current analysis aimed to determine the extent to which ML can improve on the regression based analyses that were originally carried out in the EPAD study. EPAD is unfortunately homogenous in terms of race and so it was not a factor that we could assess in our original analysis. This translated into the current paper. We appreciate however that ethnicity is a potential factor that may influence ATN status and have added this as a limitation while providing the racial distribution of the two cohorts. The exact numbers of participants, as reported in the databases, are given below (we have added summaries for both datasets in the manuscript):

o Line 183: Does the second method (following RandomForest) have a name or can be describe in addition to the citation?

Authors’ response: That citation does not refer to a further method, but also to random forest; we have changed the notation to make this more clear. We have also added the names of the other algorithms, rather than using just the names of their implementations. 

o Line 189: You explain that you derive 4 models comparing ATN groups vs controls. Is there a reason that you did this rather than use a multiclass classifier?

Authors’ response: The original purpose of the paper was to show that ML models have an advantage over more traditional ones, as the logistic regression described in Calvin et al. by using comparable dataset and the AUC metric used in the original paper. Although there are similar metrics for multiclass classifiers, they are not comparable with AUC. We are certainly interested in trying multiclass methods and we consider applying in our follow-up work, although we feel that the class imbalance (discussed in another question) and class sizes would play a role too as the smallest classes would be an even smaller portion of the total count. 

o Line 201: Please clarify what you mean by “best”

Authors’ response: these are the 100 combinations that result in the largest AUC values. We have clarified this in the text. 

RESULTS

o Most of the comparisons/models are performed on unbalanced samples and AUCs are often sensitive to class imbalance (especially in the case of normal vs AD & concomitant non-AD changes—in that case 525 vs 29). Was this considered during cross-validation (using, for example, stratified cross-validation)?

Authors’ response: Thank you for highlighting the issue of unbalanced classes. Indeed, we used stratified 5-fold cross validation and we have amended the text to indicate it. This allowed us to have stable AUC values (small standard deviation, as shown on Figure 1) for the 5 folds of the most important EPAD comparison - AD vs Normal AD biomarkers and two of the other EPAD comparisons that also have class imbalances, and also (Figure 2, bottom right pane) for the Non-AD pathologic change vs Normal AD biomarkers comparison for ADNI that has class imbalance (two of the other ADNI comparisons practically don’t have class imbalances, and one of the classes, with n=2, is too small for any reasonable analysis). On the other hand, even using stratified cross-validation resulted in unstable results for the AD and concomitant non-AD vs Normal AD biomarkers comparison on the EPAD data. This is because in addition to the severe unbalance, also the size of one of the classes is too small; that resulted in the discretization shown on Figure 1 and therefore in the instability of the results.

• Discussion/Limitations/Future Directions/Conclusion

o The Discussion does a fine job of summarizing and interpreting results from the work of the authors, but it does not contextualize the results within the large amount of work that has been and is being done both other researchers in the field. I think this needs a significant update.

Authors’ response: We agree that the original version of this section could benefit from a broader exploration of the predictive models in the field. The bulk of the work concerns amyloid positivity which is closely aligned with our Alzheimer’s disease and Alzheimer’s pathologic change models. We have amended the relevant section accordingly. 

o In Validation Procedures subsection as well as the Limitations, I think it needs to be more clearly pronounced that the validation is not EPAD-derived models used on ADNI, but features selected on EPAD were used in the training of models on ADNI.

Authors’ response: Indeed, this is a departure from the more commonly used model validation. We have added extra sentences to the Limitations section to emphasise this.

FIGURES AND TABLES

o Table 1: Could you statistically compare and report p-values for differences between, say, EPAD and ADNI as well as EPAD and ADNI-reduced? It appears there are differences in the sex, family history, APOE4, etc. makeup of the samples, but I am wondering if they are statistically significant.

Authors’ response: Below are the tables with the requested p-values. Most of the p-values are significant. The few non-significant p-values are highlighted. 

See supplementary table in the file submitted as part of the submission. 

o Table 3: It would be helpful if you could clarify how you quantitatively compared models and determined which models were similar or outperformed other models.

Answer: All comparisons were done in terms of AUC, all comparisons were done up to the second digit after the decimal point when rounding up to the fourth. We have added that clarification in the legend of that table. 

o Figures 1 and 2 are nice.

Authors’ response: Thank you.

CITATIONS

o Citations 16-18 are not informative enough for someone to know where to find the CSF cutoffs they reference.

Authors’ response: It appears that the URLs of the documents had not been added to the citations, although they appear in our reference database. We have made sure that the URLs appear in the citations and we have added the common section name, clinical values, to each of them.

Thank you for the opportunity to read your very interesting manuscript and for the contribution your work will give to the field of Alzheimer’s disease research. It was a pleasure to read your work.

Authors’ response: Thank you for the kind words and the insightful comments and suggestions, they have indeed enhanced the quality of our work.

---

## [Editor Report · Decision Letter 2]

19 Jun 2023

Identification of preclinical dementia according to ATN classification for stratified trial recruitment: A machine learning approach

PONE-D-22-31142R2

Dear Dr. Koychev,

We’re pleased to inform you that your manuscript has been judged scientifically suitable for publication and will be formally accepted for publication once it meets all outstanding technical requirements.

Kind regards,

Stephen D. Ginsberg, Ph.D.

Section Editor

PLOS ONE

---

## [Editor Report · Acceptance letter]

9 Oct 2023

PONE-D-22-31142R2 

Identification of preclinical dementia according to ATN classification for stratified trial recruitment: A machine learning approach 

Dear Dr. Koychev:

I'm pleased to inform you that your manuscript has been deemed suitable for publication in PLOS ONE. Congratulations! Your manuscript is now with our production department. 

Kind regards, 

on behalf of

Dr. Stephen D. Ginsberg 

Section Editor

PLOS ONE